# CausalCity: Complex Simulations with Agency for Causal Discovery and Reasoning

**Daniel McDuff**                                            DAMCDUFF@MICROSOFT.COM
**Yale Song**                                                 YALESONG@MICROSOFT.COM
*Microsoft, Redmond, USA*

**Jiyoung Lee**                                               LEE.J@NAVERCORP.COM
*NAVER AI Lab, South Korea*

**Vibhav Vineet**                                            VIVINEET@MICROSOFT.COM
**Sai Vemprala**                                             SAVEMPRA@MICROSOFT.COM
*Microsoft, Redmond, USA*

**Nicholas Gyde**                                            NICHOLAS.GYDE@GMAIL.COM

**Hadi Salman**                                              HADY@MIT.EDU
*MIT, Cambridge, USA*

**Kwanghoon Sohn**                                           KHSOHN@YONSEI.AC.KR
*Yonsei University, South Korea*

**Shuang Ma**                                                SHUAMA@MICROSOFT.COM
**Ashish Kapoor**                                            AKAPOOR@MICROSOFT.COM
*Microsoft, Redmond, USA*

**Editors:** Bernhard Schölkopf, Caroline Uhler and Kun Zhang

## Abstract

The ability to perform causal and counterfactual reasoning are central properties of human intelligence. Decision-making systems that can perform these types of reasoning have the potential to be more generalizable and interpretable. Simulations have helped advance the state-of-the-art in this domain, by providing the ability to systematically vary parameters (e.g., confounders) and generate examples of the outcomes in the case of counterfactual scenarios. However, simulating complex temporal causal events in multi-agent scenarios, such as those that exist in driving and vehicle navigation, is challenging. To help address this, we present a high-fidelity simulation environment that is designed for developing algorithms for causal discovery and counterfactual reasoning in the safety-critical context. A core component of our work is to introduce *agency*, such that it is simple to define and create complex scenarios using high-level definitions. The vehicles then operate with agency to complete these objectives, meaning low-level behaviors need only be controlled if necessary. We perform experiments with three state-of-the-art methods to create baselines and highlight the affordances of this environment. Finally, we highlight challenges and opportunities for future work.

**Keywords:** Causal Reasoning, Simulation

## 1. Introduction

Modern machine learning algorithms perform well on clearly defined pattern recognition tasks but still fall short *generalizing* in the ways that human intelligence can (Pearl, 2009; Schölkopf, 2019). This leads to unsatisfactory results on tasks that require extrapolation from training examples, e.g., out-of-domain recognition (Ganin and Lempitsky, 2015) and open set recognition (Scheirer et al., 2012). Causal reasoning sets human intelligence apart from pattern matching (Spelke, 2000) and enables us to answer counterfactual questions such as "what would have happened if..." Reasoning such as this is not only important in helping create learning algorithms robust to generalization but is also attractive in applications that require transparent and/or explainable decision making (e.g., safety critical scenarios including medical decision making (Richens et al., 2020) and autonomous driving (You and Han, 2020)). This is source of motivations for this work, in which we present a high-fidelity simulation of a safety-critical driving environment with vehicles that is designed for causal reasoning, we test benchmark causal inference algorithms and demonstrate how this environment can be used to systematically synthesize data to introduce complex confounders.

Discovering latent causal mechanisms and handling confounders are the key tasks in causal reasoning. Confounders refer to factors that impact both the intervention and the outcomes (Louizos et al., 2017). These factors can be "measureable" in some cases and hidden in others. If confounders are hidden then it is difficult to control for them. Ideally, we would have the ability to systematically examine the impact of many different types/classes of confounders both "hidden" and measurable whilst developing causal interference algorithms. Furthermore, we would like the ability to do so in contexts that mirror or match our real-world applications. Recent approaches to causal reasoning involve capturing causal structure and disentangling the underlying factors via an inference algorithm (e.g., neural model) (Louizos et al., 2017; Yi et al., 2019; Li et al., 2020b; Yang et al., 2020) and combining this with a graphical representations (e.g., directed acyclic graphs - DAG) to capture the underlying dynamics. These have been employed successfully to make long-term future predictions based on short observations (Li et al., 2020b).

Causal inference could make a significant impact in safety critical scenarios such as autonomous driving (Choi et al., 2019; Li et al., 2020a) where trajectory prediction is an important component. Researchers have used video and synthetic datasets (Ramanishka et al., 2018; You and Han, 2020; Kim et al., 2019; Aliakbarian et al., 2018) for analyzing causality in traffic accidents and understanding driving scenes and behaviors. However, some of these datasets are "static" (i.e., comprised of videos that cannot be changed) which means that certain counterfactual scenarios are not present and the distribution of events may be quite uneven/sparse making it difficult to learn relationships. Other datasets have limited diversity in terms of the types of events, e.g., focusing on crashes specifically (Kim et al., 2019). Our goal in this work is not to propose a new trajectory prediction algorithm but rather to illustrate how CausalCity can be used and trajectory prediction is a good task to do so.

Simulation has proven helpful as a way of investigating problems involving causal and counterfactual reasoning. The parameters of synthetic environments can be systematically controlled, thereby enabling causal relationships to be established and confounders to be introduced (Li et al., 2020b; Yi et al., 2019; Ahmed et al., 2020). However, some of this prior work has approached this via a relatively simplistic set of entities and environments (e.g., balls moving in 2D connected

. Project page: https://causalcity.github.io/

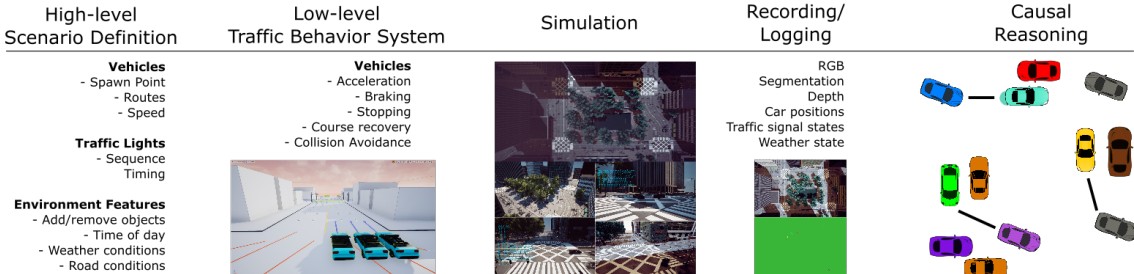

Figure 1: We present a high-fidelity simulation environment designed for experiments on causal reasoning in the safety-critical context of driving. The vehicles have agency to "decide" their low-level behaviors, which enables scenarios to be designed with simple high-level configurations. Many complex simulated scenarios can be executed with complex causal relationships. Our environment then supports the logging of rich multimodal signals during simulation for forming datasets.

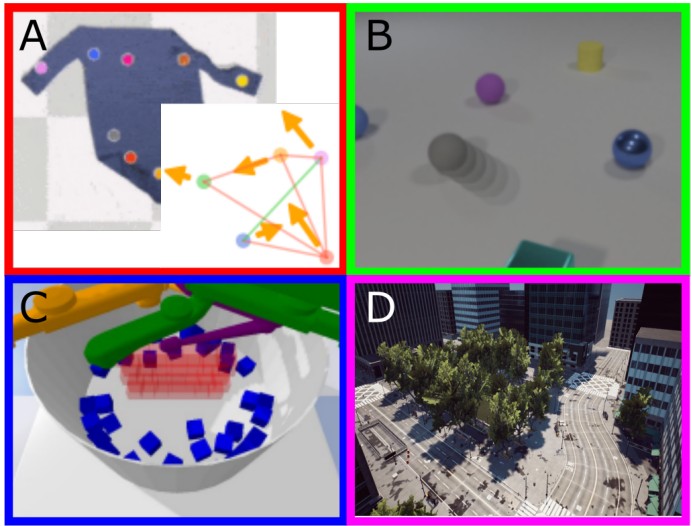

Figure 2: Simulation is a powerful tool to study causal reasoning. Here we show examples of environments used for causal reasoning. A) V-CDN (Li et al., 2020b), B) CLEVRER (Yi et al., 2019), C) CausalWorld (Ahmed et al., 2020). In contract with prior work, D) CausalCity (Ours) combines a high-fidelity visual environment with the ability to define and generate complex causal scenarios.

via rods and springs (Li et al., 2020b) or 3D objects moving on a surface and colliding (Yi et al., 2019) - see Fig. 2) with only a few variables. Other prior work has had a limited number of degrees of freedom (Ahmed et al., 2020). This leaves little room to explore, and control for, different causal relationships among entities. We posit that enabling the *agency* on each entity is crucial to creating simulation environments that reflect the nature and complexity of these types of temporal

real-world reasoning tasks. This includes scenarios where each entity makes decisions on its own while interacting with each other, e.g., pedestrians in a crowded street and cars on a busy road. What agency provides is the ability to define scenarios at a higher level, rather than specifying every single low-level action.

To this end, we introduce and publicly release a high-fidelity simulation environment, summarized in Fig. 1, with AI agent controls to create scenarios for causal reasoning. This environment reflects the real-world, safety critical scenario of driving. We want a simulation environment that enables controllable scenario generation that can be used for temporal and causal reasoning. This environment allows us to create complex scenarios including different types of confounders with relatively little effort.

To illustrate this (and as outlined in Fig. 1), our simulation engine allows the introduction of any number of vehicles, each of which is controlled at a *high-level*. Each vehicle has basic AI agency to govern *low-level* control and allow the cars to maneuver avoiding collisions, navigating corners, stopping at traffic lights, etc. The high-level controls for each vehicle allow us to define each agent's behavior in an abstract form controlling their sequence of actions (e.g., turn left at the next intersection, following that merge into the left lane etc.), their speed changes in different legs of the journey, their stopping distance behind other vehicles etc. Detailed *recording/logging* is available such that datasets can be easily created and distributed for performing *causal reasoning* experimentation. Furthermore, our simulation can be used to introduce confounders to the environment such as the time of day and the weather conditions, which can be set both changing the visual appearance of the scene but also enabling causal relationships to be introduced (for example between vehicle speed or stopping distance and the amount of water on the roads). Again, agency helps vehicles to change their behavior dynamically from the confounders. Also, traffic lights can be controlled at a low (the timing of each individual light) and high (transition timings for all the lights) levels. All these present opportunities for future work on causal reasoning. In this work we perform a set of experiments on causal discovery; however, we give other examples of how the simulation might be used in Section 6 and on our project page.

To summarize, our contributions include: 1) We present a high-fidelity simulation driving environment with vehicles (agents) that is designed for developing and testing approaches for causal reasoning. 2) We test benchmark causal inference algorithms on trajectory prediction and causal discovery tasks. 3) We demonstrate how this environment can be used to systematically synthesize data to introduce complex confounders and illustrate how these impact the performance of our baselines. 4) Our environment, a snapshot of the dataset used for analyses, and code are released with this paper (see GITHUB link on the first page). Our simulation allows for the generation of large, multimodal, complex causal datasets within the domain of vehicle navigation and we hope that it will enable researchers to tackle new research problems.

## 2. Related Work

**Causal Reasoning.** Schölkopf (2019) argues that causality and the "modeling and reasoning about interventions" can help advance machine learning as a whole and contribute to addressing some of the most challenging problems, including domain-transfer, extrapolation and other forms of generalization beyond what is explicitly observed in training datasets. These are some of the reasons that causal reasoning has received growing attention in the machine learning community.

Variational autoencoders (VAE) have been used to capture the causal structure in interactions (Louizos et al., 2017; Kipf et al., 2018) due to their ability to model uncertainty in data. The encoder can be used to estimate an unknown latent space in order to summarize the causal effects (Louizos et al., 2017) and summarize or disentangle representations of objects or events of interest from confounders (Atzmon et al., 2020). Neural Relational Inference (NRI) (Kipf et al., 2018) train an unsupervised VAE, where the latent representation captures the underlying interaction graph. The approach then learns to simultaneously capture the dynamics and infer interactions. This NRI work and others leverage a graph neural network for reconstruction (Fire and Zhu, 2017; Bhattacharya et al., 2020). Bhattacharya et al. (2020) cast causal discovery as a continuous optimization problem with differentiable constraints to find the best fitting acyclic directed mixed graph. V-CDN (Li et al., 2020b) discovers an underlying causal graph without explicit intervention in the scene and identifies interactions between entities from a short sequence of images and make long-term future predictions.

**Simulation for Causal Reasoning.** Computer graphics-based simulations have allowed researchers to explore the causality in video. PhysNet (Lerer et al., 2016) learns physics by using a 3D game engine to create small towers of wooden blocks with randomized stability. Happens (Mottaghi et al., 2016) focuses on understanding the movements of objects as a result of applying external forces to them. A large-scale dataset of forces in scenes is built by reconstructing all images in SUN RGB-D dataset (Song et al., 2015) in a physics simulator to estimate the physical movements of objects caused by external forces applied to them. Billiards (Fragkiadaki et al., 2015) learns to play a simulated billiards game, which requires planning and executing goal-specific actions in varied and unseen environments.

Johnson et al. (2017) introduced the CLEVR as a simulation engine for visual question answering (VQA). While this featured static images of objects with different shapes and colors, it was followed by the CLEVRER (Yi et al., 2019), which allows for generating objects that move and collide with one another in a 3D environment. Specifically, this enables counterfactual reasoning to be conducted. Causal World (Ahmed et al., 2020) is another recent example of simulation for causal reasoning based on a robotic manipulation benchmark. This is a 3D environment that exposes high-level variables in the causal generative model, such as properties of blocks, goals, robot links and others like gravity.

**Driving simulation.** Our proposed simulation environment is designed to be applicable to studying causal reasoning generally; however, the specific environment we chose is that of driving. The driving scenario has been used to generate synthetic data, e.g., GTACrash (Kim et al., 2019) and VIENA (Aliakbarian et al., 2018). The former involves generating a dataset for detecting car accident, whereas the latter involves generating specific actions for predicting driver maneuvers, pedestrian intentions, front car intentions, traffic rule violations, and accidents scenarios. CARNOVEL (Filos et al., 2020) is a driving benchmark specifically targeting out-of-distribution generalization using adaptive robust imitative planning (AdaRIP) and DESIRE (Lee et al., 2017) involves reasoning about scenes, context and past trajectories to predict future trajectories or locations. R2PR (Rhinehart et al., 2018) and DATF (Park et al., 2020) also approach future trajectory forecasting a task that is relevant in the context of causal reasoning in driving simulation. However, these examples are not specifically designed around the idea of causal discovery containing no counterfactual reasoning or control for confounders yet in their simulations or datasets. Also there are simulators (Shah et al., 2018; Dosovitskiy et al., 2017) that support development of autonomous cars but do not focus on causal reasoning.

Causal reasoning for autonomous driving has been attended to understand the reason of the maneuvers of other vehicles and pedestrians for escaping accidents (Ramanishka et al., 2018; You

and Han, 2020; Kim et al., 2019; Aliakbarian et al., 2018). This is a popular domain for causal analysis. Drogon is a causal reasoning framework for future trajectory forecasting (Choi et al., 2019). The authors use LiDAR data, and design a conditional prediction model to forecast goal-oriented trajectories. Finally, causal reasoning helps to reason about the behavior of vehicles as future locations conditioned on the intention. Ramanishka et al. (2018) present a dataset of 104 hours of real human driving for learning driver behavior and causal reasoning. Another benchmark for analyzing causality in traffic accident videos was presented by You et al. (2020). In this work they decompose an accident into a pair of events and analyze the cause and effect.

**Trajectory Prediction.** In temporal reasoning research, trajectory prediction is a common task (Yi et al., 2019; Li et al., 2020b), partly due to the practical utility in numerous applications (Alahi et al., 2014, 2016; Ivanovic and Pavone, 2019). In our evaluation, we use trajectory prediction as a key metric for performance and therefore it is helpful to briefly introduce work on this topic. Most of these algorithms have been developed for scenarios with a single type of agents. One such task is predicting pedestrians' future movements (Gupta et al., 2018; Vemula et al., 2018; Lisotto et al., 2019; Sadeghian et al., 2019), which is important for autonomous vehicle and robotics design. Social behaviors have been widely exploited in predicting pedestrians movements; while relevant for pedestrians, they are much less relevant for vehicles. Thus the focus on vehicle trajectory prediction has been on modeling the motion of individual agents (their past trajectory) and the surrounding environment (Lee et al., 2017; Srikanth et al., 2019; Deo and Trivedi, 2018). A notable exception is estimating lane changes on highways (Kuefler et al., 2017; Deo and Trivedi, 2018); previous efforts have tackled predicting vehicle trajectories in urban scenarios (Lee et al., 2017; Srikanth et al., 2019; Zyner et al., 2020).

Compared to single-agent scenarios, multi-agent modeling and prediction is a challenging task for control applications because agents interact with each other. Modeling dependencies between agents is especially critical in scenarios such as modelling vehicles at intersections. Previous approaches have focused on relatively sparse scenarios with only a few heterogeneous interactions. In such cases, the interaction between agents can be modelled using social forces, velocity obstacles (van den Berg et al., 2011), or linear trajectory avoidance (Pellegrini et al., 2009). When considering more complex interactions, learning-based approaches have been applied between multiple pedestrians (Alahi et al., 2016; Bartoli et al., 2018; Fernando et al., 2018; Gupta et al., 2018; Ma et al., 2017), vehicles (Chandra et al., 2019; Zhao et al., 2019; Deo and Trivedi, 2018; Lee et al., 2017; Park et al., 2018; Ding et al., 2019), and athletes (Sun et al., 2019; Zhao et al., 2019). These approaches attempt to generalize from previously observed interactions to multi-agent behavior in new situations. To perform prediction without supervision, Ehrhardt et al. (2018) learn intuitive physics from visual observations and Kipf et al. (2019) adopt contrastive learning to perform self-supervision on structured world models.

## 3. Simulation Engine

Our goal is to create a high visual-fidelity simulation environment that can be used to systematically implement complex causal relationships in realistic scenarios. For this we focus on city driving scenarios and use a set of downtown city blocks and roads with multiple four way intersections and traffic lights. Fig. 3 shows examples of the visual appearance of the simulation environment, with first person views from close to street level.

Our simulation environment – dubbed *CausalCity* – is built upon AirSim (Shah et al., 2018), which acts as a plugin for Unreal Engine and allows for obtaining training data using realistic graphics and physics simulations. Our environment contains a city block with multiple four-way intersections

and traffic lights with cars navigating through it. The environment is controlled in two primary ways. First, there is a JSON configuration file that defines a set of scenarios. Each scenario lists the vehicles that should be present, their start locations, and the high-level actions that each vehicle should take. Secondly, a python API allows scenarios to be triggered to start, parameters changed in real-time, and enables convenient logging of data as the scenarios progress. While scenarios with the same high-level definitions can be played out identically for reproducibility, it is easy to add variability by altering the number, starting points, actions or velocities of the vehicles or changing other configurations such as the timing of traffic lights, time of the day, and weather conditions.

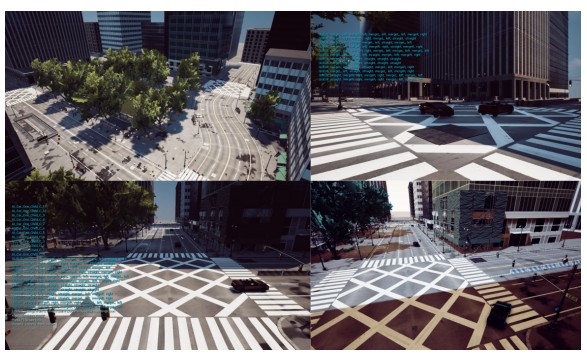

Figure 3: CausalCity simulation environment.

**Environment Features.** Our city block includes the typical elements that might be observed in such an environment (e.g., buildings, trees, lamp posts, road works, etc.) (see Fig. 3); along with well defined lanes and traffic lights that can be explicitly targeted. The objects in the environment can be easily added/removed either prior to scene simulation, or dynamically through a Python API to create different configurations of static elements.

**Vehicles.** We introduce vehicles to the scene in a systematic manner through an AI traffic module, which handles the low-level navigation controls.[1] These vehicles traverse the scene along splines (routes) that are selected via the configurable scenario file. The environment contains predefined splines running through each lane and intersection according to general traffic rules (based on right-hand drive). In the configuration file each vehicle is given a starting (spawn) point, identified by a spline ID, and a list of high-level actions to execute post-spawn. Merging actions (mergeL/mergeR) happen along lane splines and turning actions (left/right) happen at intersections. The vehicle states such as positions/velocities can be queried and obtained dynamically during scenario run-time for logging purposes. If desired, information regarding any collisions observed during the scenario can also be recorded.

**Traffic Lights.** The environment also contains traffic lights at every intersection, and the vehicles respond to these traffic signals. While keeping traffic flowing in a realistic manner, this also introduces causal connections at the intersections. The sequence and timing of these lights can be controlled during the scenario run-time. For simplicity, with our environment, we provide scripts to show how to configure the timing of lights in a sequence and how to run these asynchronously. Vehicles can be "forced" to continue driving at a red light, which can be used to simulate dangerous events and increase the likelihood of collisions.

**Environment.** Environmental factors can be modified to create variations in the visual appearance of the scene. These include introducing and controlling the strength of weather effects: rain, fog and snow; changing the time of day, and varying the wetness of the road. This allows for new parameters to be introduced as *confounders* (for example, road wetness during rain can lead to unpredictable steering behavior), and increase the variability in the observed scene in both car behavior as well as visual appearance of the scene, which makes perception tasks more challenging.

---

1. https://www.unrealengine.com/marketplace/en-US/product/arch-vis-ai-traffic-system

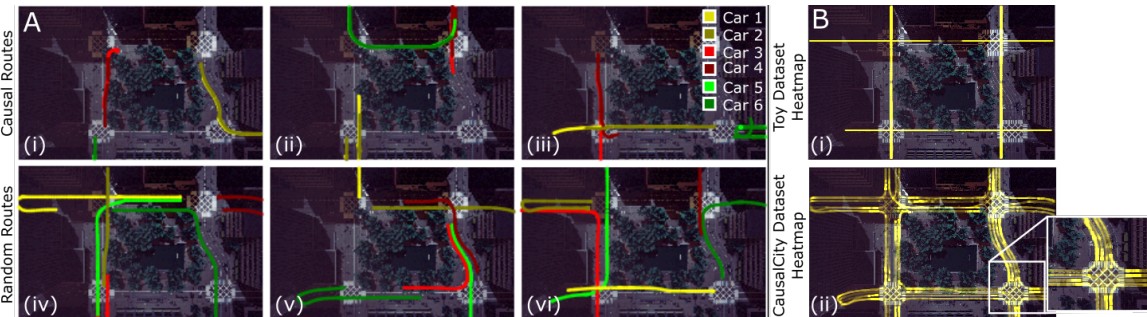

Figure 4: **Datasets.** We created two datasets, a toy dataset and our CausalCity dataset, both with cars that are connected via A) causal "leader-follower" relationships where one car follows another (A i-iii) and non-causal or random relationships (A iv-vi). B) Shows a heatmap of the paths of the vehicles in the toy dataset and CausalCity dataset. The toy dataset uses a similar city grid structure but has simplified behaviors (constant velocities, straight trajectories, etc.). The CausalCity dataset contains more realistic behaviors that introduce challenging confounders. Notice how there are longer dwell times in different lanes due to traffic patterns etc.

**Views/Cameras.** Our environment enables cameras to be placed at any location and moved during a scenario to obtain image data. This means that first person, third person, and bird's eye view perspectives are possible. For simplicity, in our first baselines presented in this paper we use a bird's eye view to visualize the scenarios. It is also possible to equip each vehicle with a camera of its own to obtain first person perspectives from the vehicles – presenting opportunities for future work.

The environment also allows for recording various modalities of data from multiple cameras for logging/visualization, such as RGB, depth and segmentation maps. In the current version, we record RGB images of the bird's eye view, as well as ground truth instance segmentation maps (generated by AirSim), where unique masks corresponding to each car in the scene are drawn for ease of use. For simplicity, our segmentation maps contain only the masks corresponding to the cars, and other scene objects are ignored.

As described in the following section, we use this framework to generate multiple scenarios, each scenario driven by the corresponding configuration setting, an example of which can be seen in Listing 1. Each scenario involves the vehicles moving through set routes, while the traffic lights and other scene variables can vary as set by the user. Data such as vehicle positions, images etc. are logged as the scenario evolves.

**Logging.** Our environment allows for rich logging of events. In the current version, for each frame we log the positions of the vehicles $(x, y, z, \sigma_x, \sigma_y, \sigma_z)$ and the state of the traffic lights (current color and duration since last change). But the positions of other objects, weather events, time of days can all also be recorded as necessary.

## 4. Dataset

To illustrate the potential of our simulation environment, we generate data and evaluate state-of-the-art causal reasoning approaches on it. Previous work has focused on causal discovery in relatively

controlled settings (e.g., balls moving in a 2D plane (Li et al., 2020b) or 3D objects colliding (Yi et al., 2019)). As we branch out to more realistic and practical scenarios (e.g., autonomous driving) we quickly encounter a number of additional complexities. One way to think about these complexities is in the form of confounders. For example, driving would be extremely difficult and unsafe without traffic signals. If we were to attempt to determine a causal relationship between the trajectory of two vehicles (e.g., is a car following an ambulance), the effect of traffic signals on their behavior could be considered a confounder.

Rather than leap directly to a context with multiple confounders, we created two versions of our data: 1) a toy dataset with causal relationships but without agency and no confounders, 2) a complex dataset created using our high fidelity simulation environment with agency (and therefore the confounders associated with it). In both cases we generated scenarios with a fixed number of vehicles (4,8,12) driving in the environment and a fixed number of causal relationships.

To introduce causal relationships between the vehicles we create "links" (or edges in the causal graph). The edges are defined as a "leader-follower" relationship in which two cars are given the same set of actions but one starts ahead of the other. In each scenario we create pairs (e.g., three pairs of six cars = three edges) of "leader-followers"; in the causal reasoning language, the leader vehicles are the *interventions* and the follower vehicles are the *outcomes*; the former causes the latter to move in certain ways. The remaining six vehicles are not causally connected to any other vehicle. This is a sparse graph (only three edges) but that is reasonable as causal relationships in the real life are often sparse. See Fig. 4A for example trajectories. This is just one example of the possible application of our simulation, see Section 6 and the project page for more examples. For simplicity in both datasets we position the camera from a bird's eye view perspective above the environment looking down. We record 150 RGB and segmentation frames for each scenario and log the position of each vehicle (6 degrees-of-freedom) at the same rate. Our dataset is available on our project page.

### 4.1. Toy Dataset

Our toy dataset uses a road layout that mimics the city block in the simulation environment. The cars do not have agency and thus move at a fixed velocities (2 pixels per frame - similar to the average speed in the CausalCity dataset), and there are no confounders such as traffic lights that influence the velocity of the vehicles. The vehicles do not collide with one another so their paths are uninterrupted. The leader vehicles start, and remain (since both have the same velocity), exactly 30 pixels in front of the follower vehicles in each pair. See our supplementary material for more examples of the trajectories of the cars in our toy dataset. We create a dataset with 4000/500/500 scenarios for the train/validation/test splits. See a heatmap in Fig. 4B that shows the average dwell time across the dataset - notice how uniform it is and contrast that with the heatmap for the CausalCity dataset.

### 4.2. CausalCity Dataset

In this dataset the cars have agency, controlled by our simulation engine, and thus have more realistic behaviors than in the toy dataset. Each vehicle has a set of five actions to complete but can drive without manually specified routes. The cars can accelerate when there is space ahead of them and reduce speed when approaching a slower moving vehicle or traffic signal. Their internal controls cause a vehicle to brake when it is approaching another vehicle. However, in some cases, if traveling fast, there may be collisions which can impact the trajectory of the cars.

Traffic lights help control the flow of traffic and also impact the velocity of vehicles regardless of causal relationships (thus they are confounders). These factors mean that even if two cars are causally linked, they will not remain a fixed distance away from each other. The "follower" vehicles may catch up with the "leader" if the "leader" is stopped at a red traffic signal, or could fall further behind it if leader makes it past a light but it turns red as the follower approaches it. See Fig. 4A and our supplementary for examples of the trajectories of the cars in our CausalCity dataset.

We observe that introducing agency to a simulation that enables highlevel scenario definitions will inevitably introduce confounders to the environment make the causal relationships more difficult to recover using the baseline algorithms (as we will see in the results). Once again, we create dataset with 4000/500/500 scenarios for the train/validation/test splits.

## 5. Experiments

We evaluate three state-of-the-art causal inference algorithms – NRI (Kipf et al., 2018), NS-DR (Yi et al., 2019) and V-CDN (Li et al., 2020b) – on both the toy and CausalCity datasets. Training was carried out on a single Nvidia P100 GPU. Each experiment typically required 10 hours of training and evaluation time.

### 5.1. Models

**NRI (Kipf et al., 2018).**    NRI is a variational autoencoder (VAE) optimized to discover a relational structure while learning the dynamical model of the underlying system. The interaction structure is explicitly modeled using a node-to-node message passing operation similar to Gilmer et al. (2017). Given sequences of locations and velocities, NRI reconstructs the original trajectory based on the predicted interaction graph. As such, the encoder learns to predict a probability distribution of edges between nodes without knowing the underlying interaction graph apriori. We leverage a recurrent neural network as a decoder to predict multiple time steps into the future and a fully-connected network as the encoder. The directed causal graph is inferred through the encoder using 100 frames of "historical" data, and then the decoder is used to predict future trajectory (up to 20 frames in our experiments) conditioned on the causal graph. We reuse predicted trajectories as inputs of the decoder to estimate the further steps. Further specifics of the implementation can be found in (Kipf et al., 2018).

**NS-DR (Yi et al., 2019).**    To accomplish causal reasoning in their work, Yi et al. (2019) used a propagation network (PropNet) (Li et al., 2019) to learn object dynamics from videos and predict object motion and collision events. We adapt this dynamics predictor model to our scenario. First, the input to the PropNet is segmentation masks of all cars in all frames of a video. These segmentation masks could be generated by popular semantic segmentation approaches. However, in our current setup, we use the segmentation masks provided by the CausalCity simulation engine, which we assume to be the upper-bound in terms of segmentation performance. Next, PropNet builds a directed graph where vertices and edges represent cars and their relationship, respectively. Each vertex encodes information about states and attributes of a car, where the states denote mask patches taken from a history of images, and the attributes denote the color of a car. We adapt our approach by assigning one unique color to each car in the scene. Finally, since we do not have collision event like in the CLEVRER dataset, our edge relationships do not contain collision state. However, including

collision events would be an interesting direction for future work in the autonomous driving context. Otherwise the implementation matches that of Yi et al. (2019) and their associated code base.

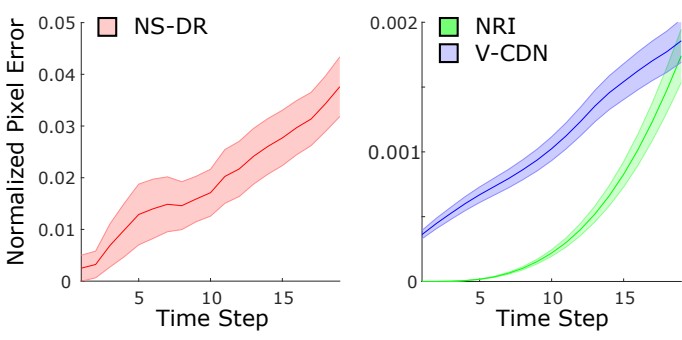

Figure 5: **Trajectory Prediction Error.** Mean square error in future trajectory prediction for eight car scenarios with two causal connection for a) NS-DR, b) NRI, b) V-CDN algorithms. Shaded regions reflect standard error. Notice the different scales on y-axes for the two plots.

**V-CDN Li et al. (2020b).** Deep graph neural networks are often used to represent underlying properties (e.g., dynamics) of physical interactions. V-CDN infers the structural causal model (SCM) from visual inputs for future prediction without supervision from the ground-truth graph structure. Li et al. (2020b) show that this can help models perform counterfactual reasoning about unseen scenarios. V-CDN consists of three parts; (1) a perception module (2) an inference module and (3) a dynamics module. Specifically, the perception module takes a sequence of images and finds keypoints, and the inference module takes these keypoints to discover a causal graph that represents the causal relationships, i.e., a physical connection in their scenario. The dynamics module is a graph recurrent network that predicts the future location of keypoints conditioned on the estimated causal graph.

We use the official implementation of Li et al. (2020b) and adapt their inference and dynamics modules, while "bypassing" the perception module for fair comparisons with other models. We take location and acceleration of the cars as input to compare the results with other baselines. Following Li et al. (2020b), for a set of cars, we construct a directed causal graph and predict the future movement of cars by conditioning on the current state and the inferred causal graph. In 5, we report the mean squared error of car locations (normalized by image dimensions from 0 to 1).

## 6. Discussion

**How do the different methods perform?** As expected we observe a gradual increase in trajectory prediction error as we attempt to predict the vehicle locations further into the future. Fig. 5 shows the normalized mean squared pixel error for time steps 1 to 20 into the future for each of the baseline methods (this corresponds to approximately 1 to 20 seconds into the future, as we sample at 1 Hz on average). The baselines show differing performance and we observe that for NS-DR the trajectory prediction errors increase more rapidly. This is consistent with previous results that found trajectory prediction to be poorer without an explicit causal discovery step (Li et al., 2020b).

**Does agency impact trajectory prediction and causal discovery?** When we contrast the performance on the toy dataset with performance on the CausalCity dataset we observe that trajectory prediction errors are larger on the CausalCity dataset as we attempt more distant future predictions. The key difference between the two datasets is the lack of *confounders* in the toy dataset (see Fig. 4B where the heatmaps contrast the trajectories and dwell times of the vehicles). In the toy dataset

the trajectories have fixed velocities and the vehicles travel on quite predictable - but less realistic - routes. In the CausalCity dataset the vehicles have agency that allows them to follow the rules of the road (e.g., drive in the correct lanes), to avoid collisions with other vehicles (i.e., brake if they are approaching another car), stop at traffic lights to reduce the risk of accidents etc.

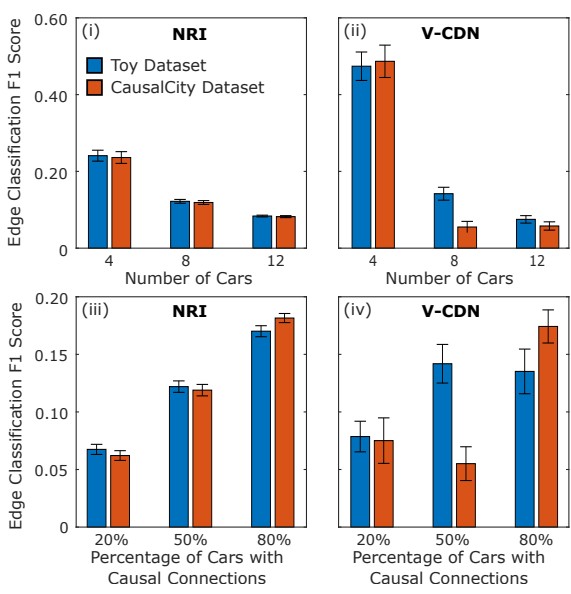

When considering the resulting trajectories of the vehicles, this adds significant - but much more realistic - confounders. The effect is a more challenging task with some room for improvement. It is clear that current state-of-the-art benchmarks struggle with trajectory prediction to some degree. This may be partially explained by the fact that causal discovery tends to be more difficult too. Fig. 6 shows the F1 scores for edge type discovery on our toy dataset and CausalCity dataset. We performed experiments varying the number of cars (4, 8, 12) and the percentage of causal connections (20%, 50%, 80%). The NS-DR method does not perform causal discovery and therefore we show the results for NRI and V-CDN. We observe that overall these are lower for the CausalCity dataset and in particular for the case with 8 cars.

**How does the number of cars and causal relationships impact results?** Discovering causal relationships is important as it can help us learn the structure of the world and make better predictions about the future. Fig. 6 shows how causal discovery performance varies with the number of vehicles and proportion of cars that have a causal connection. We observe that causal discovery becomes more difficult as the number of cars increases (holding the proportion of cars that have a causal connection constant). Greater proportions of causal connections (a less sparse causal graph) aid in causal discovery. One aspect of our task that makes causal discovery particularly difficult is how sparse the causal graph is.

Figure 6: Bar chart showing the F1 score for edge prediction. i) NRI and ii) V-CDN results for scenarios with 4, 8, 12 cars with a fixed proportion of edges (50% of cars having a causal connection). iii) NRI and iv) V-CDN results for scenarios with 20%, 50% and 80% of cars with a fixed number of cars (8). Error bars reflect standard error.

**What other tasks can CausalCity support?** We chose to demonstrate the capabilities of the CausalCity simulation on the task of causal discovery with vehicles in leader-follower style context. However, there are many other tasks in the domain of causal reasoning, discovery and counterfactual reasoning that the simulation could be used for. For example, our simulation enables a "hero" vehicle to be used to create targeted interventions in the scene and such a method could be used to test the ability for algorithms to reason counterfactually (i.e., what would have happened if the hero vehicle did not stop at the traffic signal?). As this is a simulation it is possible to generate a scene with the same initial starting conditions but to strategically intervene with a specific action at a specific moment. We have included an example of how to conduct this type of experiment in our repo.

## 7. Broader Impacts

Causal reasoning presents promising opportunities for machine learning. Specifically, causal discovery and reasoning could help create models that are more explainable. Therefore, tools that help advance this understanding will be valuable to the research community. However, we must acknowledge some limitations of our system. Our simulation environment is designed around the task of driving; however, this does not mean that a system trained on these data will be appropriate for real-world applications. The scenarios created in our simulation are complex and do have reasonable visual fidelity, but the are still a long way from simulating realistic behavior of drivers. We are adding pedestrians to the simulation engine but it does not currently feature animals (e.g., birds) which are another commonly occurring element in everyday environments. This environment was designed for experimentation, specifically in the domain of causal discovery; generalization to real-world tasks - especially safety critical ones like driving - would require greater testing on real-world data.

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
