# OpenReview forum: "CausalCity: Complex Simulations with Agency for Causal Discovery and Reasoning"
_cclear.cc/CLeaR/2022/Conference — CLeaR 2022 Poster_

### Official Review · Reviewer_di38 · 2021-11-19

**Confidence:** 3
**Overall Score:** 6

**Main Review:**

One important limitation when developing causal inference algorithms is the lack of real-world datasets for which the ground truth causal relations are known, which is why I think the CausalCity simulation environment is a welcome addition. It seems to me that the authors have managed to design an engine that can simulate relatively complex and reasonably realistic scenarios. The most novel aspect of this simulation, as far as I understand, is the addition of low-level _agency_ to the vehicle objects, which allows the simulation designer to focus on the high-level actions that the vehicles are supposed to take (merging lanes, turning). The paper is very well written and virtually error-free.

As the authors themselves admit, this environment is limited to driving tasks. Another limitation of the present work is that running the simulation code seems to require a bit of toil. The GitHub page has a rather nice presentation, but I would have liked it if the authors had included a dataset to be used directly in the code repository. I could not find such a dataset there.

Overall, I think the authors have proposed a handy simulation environment that can be used to create more realistic synthetic datasets, which in turn can help toward the design of causal inference algorithms.

Minor comments:
- page 9, Subsection 4.2: "highlevel" -> "high-level", "we create dataset" -> "we create datasets"
- page 10, last line before Discussion section: "In 5" -> "In Fig. 5"

**Summary:**

The authors introduce a high-fidelity simulation environment for driving scenarios (CausalCity) that can produce benchmark datasets for evaluating causal discovery and counterfactual reasoning algorithms. The vehicle objects have agency in navigating the environment (e.g., avoiding collisions, stopping at traffic lights), which enables the researcher to focus on defining high-level actions (interventions) to be studied.

---

> ### Author Response · Authors · 2021-12-01
> **Response.**
>
> Make the links to the dataset easier to find/add them to the Git repo:
>
> We had links to the datasets on the project page (https://causalcity.github.io/dataset.html) these are hosted in a cloud storage account as they are too large to put in the Github repo. However, we will add the link to the Github readme, as we think this would help - https://github.com/causalcity/causalcity.github.io/blob/main/README.md.
>
>
> Minor comments:
>
> Thank you for bringing those typos to our attention.  We have corrected them in the manuscript.

---

> > ### Comment · Reviewer_di38 · 2021-12-10
> > **Acknowledgment**
> >
> > Thank you to the authors for addressing my few concerns.

---

### Official Review · Reviewer_VwpM · 2021-11-22

**Confidence:** 3
**Overall Score:** 8

**Main Review:**

This paper builds on AirSim, a simulator for drones and cars in a photo-realistic setting, and creates a driving environment with multiple autonomous agents controlled by high-level directives. The main contribution is the agency of the vehicles and a concrete implementation of the AirSim software in a causal learning setting, along with some baselines. This is an original simulation environment and it differs from others by its ability to control different aspects of the simulation, thus enabling counterfactual evaluation. The related works are discussed and compared clearly.

One of the main challenges in causal learning is the lack of a comprehensive and popular dataset on which causal learning algorithms are all tested to measure progress in the field. One of the roadblocks to such a dataset is that contrary to classic machine learning, causal learning is all about interventions and counterfactual reasoning. A static dataset can hardly cover all the possible scenarios, making simulation a powerful tool to test methods on. However, complex and realistic simulations are hard to create, therefore many simulated environments are often simplistic synthetic examples that can’t guarantee good performance in real-life scenarios. This work is important in that it tries to bridge this gap, and provides a realistic setting all while maintaining the ability to control many aspects of the simulation, for example, confounders.

The website presenting the simulation is well done and the examples and figures give a good idea of what the simulation is about. However, I found the code of the simulation itself to be a little under-documented and not very clear. For instance, I have not found anywhere the JSON files that form the basis of the scenarios so I have no idea how they are structured or what is expected as input. The paper talks about controlling weather and time of day etc… but it is not clear where one should start to achieve that. I realize that most of the code to do this lies in the AirSim documentation. However, it would be nice to have pointers to the commands that allow changes related to confounders and vehicle behavior. Of course, the code in itself is supplementary material to the paper, and I am sure that it is a work in progress and that the documentation could be clarified even by other people in the community and this could serve as a stepping stone to a more user-friendly simulation platform. I have not tried to reproduce the simulation so I cannot speak to how straightforward it is.

The paper itself is well organized and I have found it quite a pleasure to read. The motivation is very clear and the different simulations are well illustrated. Overall the main idea of the paper resonates easily with the interested reader.

I am excited to play around with it as I think it has some potential! I hope the authors will make the code more accessible to people unfamiliar with AirSim.


**Summary:**

This paper proposes a new high-fidelity simulation environment built on the AirSim software. This simulation aims to provide a true-to-life scenario, with a safety-critical context, for causal learning algorithms. They showcase the use of their simulation on the tasks of trajectory prediction and causal discovery and create a baseline with three existing algorithms. The simulation revolves around vehicles driving in an intersection with various controllable factors such as camera view, traffic lights, weather, time of day, buildings, lamps, road works, etc… Most importantly, the simulation features vehicles equipped with an AI agent that can be controlled through high-level directives such as turning at the intersection, merging, etc… The ability to control all these factors enables evaluating models on counterfactual scenarios, making the simulation much richer than others that generate a static dataset.

---

> ### Author Response · Authors · 2021-12-01
> **Response.**
>
> Thank you for the helpful comments and encouraging review.
>
> Provide more examples of how to create the JSON config files:
>
> Thank you for bringing this to our attention. We have added a script to the Github repo (https://github.com/causalcity/causalcity.github.io/blob/main/code/scenarioGenerator.py) showing how to create scenarios and have provided a greater explanation of how to construct the JSON files in the Github repo readme (https://github.com/causalcity/causalcity.github.io).
>
>
> Provide examples of how to control confounder variables:
>
> The reviewer is correct: weather, road conditions, traffic signals and time of day could all be examples of confounders in the simulation.  We will add examples of how to control these confounder variables directly into our documentation. Some of the confounders will only directly impact the visual appearance of the scene (e.g., time of day), while others will directly impact the movement of the vehicle (e.g., traffic signals). We will clarify which applies in each case.

---

> > ### Comment · Reviewer_VwpM · 2021-12-15
> > **Acknowldgement**
> >
> > I thank the authors for addressing my concerns!

---

### Official Review · Reviewer_Xryg · 2021-11-27

**Confidence:** 4
**Overall Score:** 6

**Main Review:**

This paper presents a novel and useful simulation environment that can be used to evaluate certain classes of causal models. Evaluation in causal learning and reasoning is complex, largely because causal models are intended to evaluate the effects of actions, making many static data sets insufficient for evaluation. In addition, algorithms for causal learning and reasoning are increasingly designed for tasks that go beyond simple estimation of average treatment effects.

That said, several aspects of the presentation could be substantially improved. First, it takes two pages before the introduction actually identifies the key contribution of the paper (“To this end…”). This should appear much earlier, preferably on p. 1.  While Figure 1 presents some details earlier, that figure is not intelligible until readers get some more general description of CausalCity.

Second, given that CausalCity is built on top of AirSim (and already very capable simulation environment), the authors would substantially improve the paper by highlighting the unique features that they added to enable CausalCity to be useful for R&D in causal inference.  Such features should be the central topics addressed by each subsection rather than the current focus on the specifics of a driving simulator (e.g., “environment features”, "vehicles").

Third, most simulators would appear to provide opportunities for doing many of the things listed as advantages of CausalCity. The paper should identify the unique advantages of CausalCity for research in causal inference that are not provided by other simulators. While it is beneficial to research on causal inference to have “another simulator”, it would be useful to highlight why this particular simulator will help research in causal inference more than others.

Finally, the discussion of a model (NS-DR) begins by discussing a data set. It would be far more useful to begin by discussing the model itself. This is what readers will expect.

**Summary:**

A useful contribution that could be better presented

---

> ### Author Response · Authors · 2021-12-01
> **Response**
>
> Establish what CausalCity provides above and beyond AirSim:
>
> We agree that simulators in general are useful for causal analyses, and AirSim could be used in this way directly depending on the task.  However, we have added specific features that enable more elaborate and complex modeling, that we argue makes it easier to create more realistic scenarios. The key unique features of CausalCity are: 1) the ability to define routes for any number of vehicles using spawn and waypoints and actions at those points (e.g., turning, changing lanes, stopping, etc.) through an easily configurable interface, 2) the vehicles have basic agency such that it is not necessary to define the complete trajectory of every vehicle – what this enables is the ability to create much more complex scenarios more easily and focus on high-level behaviors. Of course, the expected low level causal relationships that one would expect still exist, such as the consequences if two cars collide. In contrast, in AirSim only low-level controls exist and it would be necessary to define the complete trajectory of every vehicle at every time point, which is laborious.
>
> Highlight the contributions earlier:
>
> We appreciate the suggestion regarding moving the contributions earlier in the text. We agree that this would help. We will bring a summary of the main points to the first paragraph to help situate the paper and provide more context for Fig. 1.
>
> Remove the description of the data from the discussion of NS-DR:
>
> This is another good suggestion.  We will remove the description of the CLEVRER data, as that is not particularly relevant to our experiments and focus on the explanation of the method.

---

### Author Response · Authors · 2021-12-01
**Appreciation.**

We would like to thank the reviewers for their positive and constructive feedback.  We are glad they felt that the CausalCity project provided a novel, useful and welcome addition to the community. Below we have responded to each reviewer with the changes we will make to the paper to address their comments.

---

### Decision · Program_Chairs · 2022-01-12

**Decision:**

Accept (Poster)

**Comment:**

The reviewers agree that the CausalCity simulation engine provides a useful testbed for causal inference methods that combines an ability to simulate counterfactual scenarios with appropriate complexity to feel like a non-trivial benchmark. At the same time, the configuration interface provided by the simulator provides a usable way to customize that complexity. I agree with Reviewer Xryg's comment that the paper presentation could be improved in the intro and beyond in emphasizing the particular contributions of this work. This may require some substantial refactoring of the presentation, but because this is mostly a non-technical issue, I don't think such revisions would require further review.